# Cognitive Impulsivity in Anorexia Nervosa in Correlation with Eating and Obsessive Symptoms: A Comparison with Healthy Controls

**DOI:** 10.3390/nu16081156

**Published:** 2024-04-13

**Authors:** Francesco Bevione, Matteo Martini, Federica Toppino, Paola Longo, Giovanni Abbate-Daga, Annalisa Brustolin, Matteo Panero

**Affiliations:** Eating Disorders Unit, Department of Neuroscience “Rita Levi Montalcini”, University of Turin, via Cherasco 15, 10126 Turin, Italy; francesco.bevione@unito.it (F.B.); matteo.martini@unito.it (M.M.); federica.toppino@unito.it (F.T.); paola.longo@unito.it (P.L.); annalisa.brustolin@gmail.com (A.B.); matteo.panero@unito.it (M.P.)

**Keywords:** eating disorders, anorexia nervosa, impulsivity, obsessive symptoms, eating psychopathology

## Abstract

Impulsivity in eating disorders (ED) has been historically focused on bingeing–purging symptoms, evidencing lower levels in restricting subtypes. In the recent literature, obsessive–compulsive disorder (OCD) has been described as characterized by high cognitive impulsivity. This specific impulsivity factor has been rarely studied in anorexia nervosa (AN). In this study, 53 inpatients with anorexia nervosa and 59 healthy controls completed the following questionnaires: the Barratt Impulsiveness Scale (BIS-11), the Obsessive–Compulsive Inventory (OCI), the Eating Disorders Inventory—2 (EDI–2), the State–Trait Anxiety Inventory (STAI), and the Beck Depression Inventory (BDI). AN individuals showed significantly increased levels of cognitive instability but no difference in global score and other subscales of impulsivity compared to the healthy controls. Among AN individuals, cognitive instability emerged as being associated with the global score and obsession subscale of the OCI. It was also significantly associated with interoceptive awareness and impulse regulation. Cognitive instability was the main predictor of obsessive thoughts and behaviors in AN. Our study supports the hypothesis of AN as being characterized by high cognitive instability and adds the result that the cognitive domain of impulsivity may be associated with the presence of obsessive symptoms, specifically obsessive thoughts.

## 1. Introduction

Eating disorders (EDs) are important psychiatric disorders with heavy impairment of quality of life and commonly manifest with a chronic or relapsing trajectory [1,2,3]. 

Among psychological features that are thought to foster or be strengthened by the onset of EDs, impulsivity is a neuropsychological construct that has been associated with all of them, specifically with those eating disorders characterized by bingeing and purging symptoms [4], and it is sometimes conceptualized as a shared “neurocognitive endophenotype” [5]. 

Impulsivity can be defined and described according to many different models, sometimes leading to contradictory findings [6]. The model that historically received the most agreement is the impulsivity model proposed by Barratt [7], which conceives impulsivity as a multi-faceted entity comprising six first-order factors (attention, motor impulsiveness, self-control, cognitive complexity, perseverance, and cognitive instability) that combined constitute the three second-order factors (attentional impulsiveness, motor impulsiveness, and non-planning impulsiveness). Impulsivity according to Barratt’s model can be evaluated through the Barratt Impulsiveness Scale-11 (BIS-11), a 30-item questionnaire with three subscales corresponding to the three second-order factors (cognitive, motor, and non-planning) [7]. Another model widely recognized is the five personality dimensions impulsivity model (UPPS) proposed by Whiteside and Lynam [8], in which five personality dimensions are related to impulsive behaviors: positive and negative urgency, lack of premeditation, lack of perseverance, and sensation seeking. Finally, the reinforcement sensitivity theory (RST) proposes that the personality dimensions linked to impulsivity are mediated by two biologically based motivational systems: the behavioral approach system (BAS), regulating the response to appetitive stimuli, and the behavioral inhibition system (BIS), regulating the inhibited response to aversive stimuli [9].

Zooming in on EDs, research exploring impulsivity has been historically focused on bulimic spectrum disorders, namely bulimia nervosa (BN), binge eating disorder (BED), and the binge eating/purging subtype of anorexia nervosa (AN-BP) [4]. In a study, AN-BP and BED individuals showed significantly lower reward delay discounting and higher positive and negative urgency levels of impulsivity than AN-R individuals [10]. In another study, AN-BP and BN individuals achieved higher impulsivity scores than AN-R individuals [11]. Notwithstanding a more pronounced trend in the bulimic spectrum, all individuals with diagnoses display higher impulsivity compared to healthy controls [4,10,11,12,13].

Besides EDs, impulsivity dimensions are common to many other psychiatric disorders. The BIS-11 scale has been widely adopted, with it describing increased impulsivity among borderline personality disorder [14,15,16], ADHD [17], antisocial personality disorder [14], alcohol and substance use disorder [18], bipolar disorder [17], obsessive–compulsive disorder (OCD) [19,20], and forensic individuals [21]. The most interesting findings are indeed those associated with OCD, historically considered a disorder characterized by inhibited and hypercontrolled rather than explosive behavior. A study [22] in OCD individuals compared to healthy controls found significant differences only in attentional impulsivity but not in any other neurocognitive impulsivity domain [22]. Another study [23] showed that emotion-related impulsivity, but not non-emotion-related impulsivity, was associated with greater severity of obsessive symptomatology and that impulsive cognitive responses to emotion were associated with obsession severity.

Although OCD is portrayed as characterized by low impulsivity, more recent clinical, neuropsychological, and neuroimaging studies have challenged this idea. The presence of increased impulsivity [24], risky decision-making [25], and reward system dysfunction [26] conflict with the stereotypical OCD portrait. In a study [20], OCD patients had significantly higher BIS-11 scores than controls, in particular on the cognitive subscales. As AN has often been compared to OCD, a deeper study of impulsivity in AN would be interesting to investigate as AN could, similarly to OCD, present high levels of impulsivity, even when the stereotype and the clinical descriptions of AN are generally associated with lower impulsivity when compared with other ED diagnoses and healthy controls. A review of the literature [13] summarized that bingeing and restricting behaviors may be seen as lying on opposite ends of a spectrum of impulsive behaviors but also notes that AN individuals showed higher impulsivity in continuous performance tasks (CPTs) in the go/no go task. This may indicate the need for a more complex analysis of impulsivity in AN. According to a recent study [19], ED and OCD might be characterized by increased levels of impulsivity, and a lack of inhibitory control may represent a specific behavioral deficit in OCD but not in ED. In this perspective, further studies may benefit from analyzing the interplay between eating and obsessive symptomatology and investigating whether or not the presence of impulsivity plays a specific role. A study that assessed the response inhibition and cognitive and behavioral impulsivity in bulimic spectrum ED and OCD persons [19] evidenced that both groups of individuals reported higher levels of cognitive impulsivity, none of them showed significant differences in behavioral impulsivity, and only the OCD group showed an increased response inhibition time compared to healthy controls. In another study [27], AN-R individuals showed reduced impulsivity levels and poorer decision-making performance compared to those with AN-BP. Impaired decision-making and working memory in EDs, specifically in AN-R, have been described as cognitive impairments that may be connected to impulsivity [28,29,30,31,32]. Research assessing impulsivity in EDs has been historically focused on the bulimic spectrum [33]. Schaumberg et al. [34] evidenced how, in BN, disordered behavior is specifically related to a particular impulsivity dimension. For example, binge eating is associated with risk-taking, laxative misuse is associated with impulsive spending, fasting is associated with difficulty concentrating, and vomiting is associated with long-term planning difficulties. Impulsivity in relation to restrictive eating behaviors has been much less explored in research. A more detailed analysis of impulsivity in AN may reveal different facets of this neuropsychological characteristic. This is a field of research where further studies are needed. Our paper aims to expand the number of studies and investigate the link between eating and obsessive symptoms, accounting for the role of impulsivity [35]. 

Given these premises, the study’s primary aim was to compare AN individuals with healthy controls regarding impulsivity. Secondly, we aimed to explore which psychological dimensions showed the most significant associations with impulsivity among AN individuals, evaluating eating and general psychopathology and obsessive thoughts and behaviors. Finally, we aimed to assess the main predictors of obsessive thoughts and behaviors among AN individuals when taking into account impulsivity.

Based on previous findings in the literature, we expect that AN individuals present increased impulsivity levels compared to healthy controls, that impulsivity is associated with bulimic symptomatology, and that impulsivity predicts a higher presence of obsessive thoughts and behaviors. Furthermore, we expect that in AN individuals, attentional rather than other (motor and non-planning) dimensions of impulsivity have higher scores.

## 2. Methods

### 2.1. Participants

We recruited 53 inpatients with AN at the Eating Disorders Center of the “Città della Salute e della Scienza” subsequently admitted to the Eating Disorder Center of the University of Turin, Italy, and 59 healthy controls (HCs). The patients comprised 41 with the restricting AN (AN-R) subtype and 12 with the binging–purging subtype (AN-BP). All were diagnosed with AN according to DSM-5 [36] by an experienced psychiatrist following the Structured Clinical Interview for DSM-5 [37]. The following inclusion criteria were set: (a) age range: 18–65 years old and (b) diagnosis of AN according to the Structured Clinical Interview for DSM-5 [37]. The exclusion criteria were: (a) alcohol or substance dependence; (b) medical comorbidities (e.g., epilepsy or diabetes); (c) history of cranial trauma with loss of consciousness; and (d) psychotic disorders.

Seventeen individuals (32%) were suffering from other psychiatric illnesses (fifteen were diagnosed with major depression disorder and two were diagnosed with anxiety disorder); twenty-two (41%) individuals were under psychotropic medications (antidepressants and anxiolytics).

Fifty-nine gender-matched HCs were recruited from visiting students and collaborators via the snowball sampling method; they were then interviewed in person to measure their BMI and to ascertain the following inclusion criteria: (1) no lifetime history of mental disorders according to DSM-5 criteria; (2) no use of medications; (3) no current or lifetime organic illness as assessed per clinical interview; and (4) age > 18 and <55 years old.

No one refused to complete the questionnaires. All participants voluntarily agreed to be involved in our study, declaring this through written informed consent according to the ethical committee of our institution, which also approved the present study under registration number 00295/2021 of 3 June 2021.

### 2.2. Procedure and Measures

Patients were measured in height and weight by trained nursing personnel. After that, the participants were interviewed by an experienced psychiatrist who collected clinical and demographic data. Specifically, for each patient, information was collected regarding gender and illness onset. 

All participants were asked to complete the following self-report questionnaires:Barratt Impulsiveness Scale version 11 (BIS-11; [7]). The Italian version of this questionnaire was utilized to assess impulsivity. This tool consists of a 30-item questionnaire with six first-order factors (attention, motor impulsiveness, self-control, cognitive complexity, perseverance, and cognitive instability) that combined constitute the three second-order factors (attentional impulsiveness, motor impulsiveness, and non-planning impulsiveness), resulting in a global score. Higher scores correspond to higher impulsivity, with good internal consistency of the Italian validation (Cronbach’s alpha = 0.79; test–retest reliability = 0.89; [14]).Obsessive–Compulsive Inventory (OCI; [38]). The Italian version of this questionnaire evaluates obsessive thoughts and behaviors, consisting of 42 items comprising 7 subscales: washing, checking, doubting, ordering, obsessing, hoarding, and mental neutralizing. The Italian version showed good internal consistency (Cronbach’s alpha > 0.90; [39]).Eating Disorders Inventory—2 (EDI-; [40]). This tool, used in the Italian version, was specific to derive an indicator of eating psychopathology. This questionnaire is organized into 11 subscales: drive for thinness, bulimia, body dissatisfaction, ineffectiveness, perfectionism, interpersonal distrust, interoceptive awareness, maturity fears, asceticism, impulse regulation, and social insecurity. Higher scores indicate more severe eating disorder symptoms, with good internal consistency of the Italian validation (Cronbach’s alpha > 0.90; [41]).State–Trait Anxiety Inventory (STAI; [42]). This tool, in the Italian version, was utilized to assess anxiety levels. Specifically, 20 items evaluate anxiety in the present moment (“state anxiety”), and 20 items assess basal levels of anxiety (“trait anxiety”). Higher scores indicate increased levels of anxiety. The internal consistency was good, with Cronbach’s alpha values between 0.86 and 0.95 [43].Beck Depression Inventory 1 (BDI; [44]). This questionnaire was employed, in the Italian version, to assess depression levels. It consists of 21 items for a maximum score of 63, where higher scores correspond to greater depression with good internal consistency (Cronbach’s alpha = 0.87; test-retest reliability > 0.70; [45]).

### 2.3. Statistical Analysis

All statistical analyses were performed utilizing SPSS Statistics 28.0 Software (IBM Corp, Armonk, NY, USA).

Firstly, we compared the age and body mass index (BMI) in AN individuals and HCs to ascertain bias that could undermine the interpretability of the results.

Secondly, we ran an independent-sample *t*-test to compare AN individuals and HCs on BIS-11 scores to assess whether differences in impulsivity dimensions were detected between the two groups.

Subsequently, we carried out a Pearson’s correlation to explore which dimensions correlate the most with impulsivity in the clinical sample only and in the healthy control group. Overconservative Bonferroni–Holm correction for multiple comparisons was applied. Finally, through linear regression, we analyzed the main predictors of obsessive symptoms in the AN sample and included in the model the following variables: age, BMI, BIS-11 cognitive instability, EDI–2 body dissatisfaction, EDI–2 bulimia, EDI–2 drive for thinness, STAI state anxiety, STAI trait anxiety, and the BDI. These variables were chosen to include the three eating symptomatology scales of EDI-2, the scales of anxious and depressive comorbidities, and the BIS-11 scale which has proven to be significant in the *t*-test.

## 3. Results

The clinical sample and the control group had similar ages (mean AN = 22.54; mean HCs = 23.42; *p* = 0.171) and significantly different BMIs (mean AN = 15.69; mean HCs = 20.28; *p* = 0.001). Patients comprising the AN sample were severely underweight, with a long duration of illness (mean years of illness = 6.0; SD = 4.41). In the HC group, the mean EDI–2 scores were below the cut-offs for EDs. The mean values and standard deviations of BMI, years of illness, the EDI–2, the BDI, and the STAI are detailed in Appendix A.

### 3.1. Differences in Impulsivity Dimensions between AN Individuals and Healthy Controls

Impulsivity dimensions were assessed in AN individuals and HCs by utilizing the BIS-11. The participants’ scores are detailed in Table 1. 

The only factor that proved to be significantly altered in AN was the cognitive instability score of the BIS-11, a measure of cognitive impulsivity. No other factors emerged as being significantly different between the two groups.

### 3.2. Correlations of Total Impulsivity and Cognitive Instability in AN Individuals

In the AN individual group, total impulsivity showed associations with the bulimia, ineffectiveness, impulse regulation, and interoceptive awareness scores of the EDI–2, the BDI, and the obsessing subscale of the OCI. Notably, none of these variables remained significant after the Bonferroni–Holm correction was applied.

Cognitive instability was associated with the global score, obsessing, mental neutralizing, hoarding, doubting, ordering, washing, and checking subscales of the OCI, the interoceptive awareness, impulse regulation, social insecurity, ineffectiveness, and interpersonal distrust of the EDI–2, the depressive symptoms as measured with the BDI, the STAI trait anxiety, and years of illness. Moreover, a significant inverse association emerged with age. The associations of cognitive instability that remained significant after Bonferroni–Holm correction was applied were with the global score, and obsessing subscale of the OCI and the interoceptive awareness and impulse regulation subscales of the EDI–2. 

Data are detailed in Table 2. 

In the control group, there were no associations between the BIS-11 global score and the OCI, STAI, BDI, BMI, and age; meanwhile, the scores of cognitive instability showed a correlation with the washing, doubting, hoarding, global score, and obsessing scales of the OCI, with only the latter result being significant after Bonferroni–Holm correction. Data are shown in Appendix A. 

### 3.3. Linear Regression of Obsessive Thoughts and Behaviors in AN Individuals

We performed linear regression to derive which dimensions predict the most obsessive thoughts and behaviors in AN individuals. The results are detailed in Table 3. 

The strongest predictor was the cognitive instability score of the BIS-11. Other significant predictors proved to be body dissatisfaction and, inversely, the bulimia scores of the EDI–2. It is noteworthy that depressive and anxiety symptoms assessed with the BDI and the STAI emerged as being non-significant. 

Also, in the control group, cognitive instability, with trait anxiety and bulimia, were predictors of obsessive thoughts and behaviors. The regression model for the HCs is shown in Appendix A. 

## 4. Discussion

This study aimed to explore impulsivity and obsessive thoughts and behaviors in AN by analyzing the differences in impulsivity between AN individuals and healthy controls, the main associations between impulsivity and psychopathological characteristics, and the predictors of obsessive symptoms in AN. Three principal findings emerged. Firstly, AN individuals showed statistically significantly increased cognitive impulsivity levels but no difference in global score and other impulsivity subscales compared to the healthy controls. Secondly, in AN individuals, total impulsivity emerged as being associated with the obsessing subscale of the OCI; cognitive instability proved to be associated with the OCI global score and obsessing subscale of the OCI and the interoceptive awareness and impulse regulation subscales of the EDI–2. Finally, cognitive instability was the strongest predictor of obsessive thoughts and behaviors in AN individuals. Other significant predictors were the body dissatisfaction and bulimia subscales of the EDI–2, while depressive and anxiety symptoms proved to be non-significant.

### 4.1. Impulsivity and AN

Our study’s results indicate that cognitive impulsivity is elevated in AN individuals. AN individuals were shown to differ from the healthy controls only in cognitive instability, while not varying in global and any other subscale of impulsivity (attentional, motor, and non-planning). These results add information with regard to the role of cognitive impulsivity in eating disorders [23].

### 4.2. The Relationship between Impulsivity and Psychological Characteristics in AN

In the clinical sample, the associations between cognitive instability and the OCI subscales were the global score and obsessing subscale. Furthermore, the association between cognitive instability and all other subscales of the OCI was present but did not survive after Bonferroni–Holm correction, also suggesting a broader connection between cognitive impulsivity and obsessive symptoms [4,46]. Finally, considering global impulsivity, obsessive thoughts (i.e., the obsessing subscale of the OCI) emerged as being associated. Our data could support the relationship between impulsivity, especially in the cognitive domains, and obsessionality in AN.

Regarding the relationship between impulsivity and eating behavior, global impulsivity was not associated with any of the EDI-2 subscales. Conversely, cognitive impulsivity (i.e., the cognitive instability score of the BIS-11) was correlated with interoceptive awareness and impulse regulation. Hence, these data suggest that general impulsivity is not associated with eating symptoms in AN but that cognitive instability is related to AN’s core characteristics of interoceptive disturbances and the presence of obsessive thoughts. 

Our data confirm the stereotypical notion that global impulsivity (specifically motor and non-planning) is not associated with AN, but they suggest the presence of a role for higher attentional impulsivity, specifically cognitive instability in AN, associated with obsessive thoughts and behaviors.

In the correlation analysis, the association between the cognitive instability, interpersonal distrust, and social insecurity dimensions of the EDI–2 did not survive after correction for multiple comparisons. The finding of an association between cognitive instability and interpersonal distrust and social insecurity would have indicated the existence of mental processes characterized by cognitive impulsivity connected to psychosocial difficulties in interpersonal relationships. Further studies with a larger sample size may help in linking avoidance behaviors typical of AN, negative emotions that foster avoidance behaviors (like shame and guilt) [47], and the presence of cognitive impulsivity [4]. 

Both the association between total impulsivity and cognitive instability and depressive symptomatology did not remain after the Bonferroni–Holm correction. Previous results stated that the intensity of depression influences the intensity of obsessive–compulsive symptoms in EDs [48,49,50]. This could be a further difficulty factor for treatments and resistance to treatment [51,52,53,54,55,56]. In the control group, cognitive instability was also found to be associated with the obsession subscale of the OCI. This association therefore might not be specific to AN and thus not associated with being underweight or having a psychiatric disorder. Therefore, this result deserves further investigation through samples that are more representative of the general population.

### 4.3. The Relationship between Obsessive Symptoms and Psychological Characteristics in AN

Finally, the linear regression showed cognitive impulsivity as the principal predictor of obsessive thoughts and behaviors in the AN sample. This finding further stresses the importance of the interplay between cognitive impulsivity and obsessionality. Other significant predictor of obsessive symptoms were body dissatisfaction and negatively the bulimia scale of EDI-2. Notably, depressive and anxiety symptoms resulted in non-significant predictors. Even though, in the literature, there is evidence of a worsening of obsessional thoughts with depression [47,48,49], obsessionality in AN may be independent of depressive comorbidity. As cognitive instability also predicted OCD symptoms in the healthy controls, our results suggest that the relationship between cognitive impulsivity and OCD symptoms is not specific to AN but generalizable to other populations.

In conclusion, our study supports the hypothesis that obsessive thoughts and behaviors are associated with cognitive impulsivity in AN. Our data add the insight that cognitive instability is associated with obsessive symptoms, suggesting that cognitive impulsivity may play a similar role in AN and OCD. 

Some limitations must be acknowledged. Firstly, the small size of our sample did not allow for any comparison between the diagnostic subtypes. From this perspective, it would be especially interesting to compare AN-R and AN-BP individuals to allow for a direct comparison. The aim of future studies would be to enlarge the clinical sample to comprise BN patients to allow for a comparison between different EDs. Secondly, a major limitation is that the BIS-11 factor structure is debated [23]; we chose to maintain the BIS-11 because it has been used in research into OCD, and the objective of our study was to offer a comparison between OCD and AN [57]. Third, we only utilized self-report measures. Fourth, the cross-sectional study design did not allow us to conclude any causal relation but only statistical associations between variables. Fifth, the data were not controlled for age or body mass index. Finally, in this study, the presence of behavioral or mental inhibition was not assessed; further research may need to assess this factor independently.

## 5. Conclusions and Practical Implications

The present study’s findings suggest that cognitive impulsivity is a factor to be addressed in the context of research and treatment for AN [58]. The role of cognitive impulsivity may be suspected especially when in the presence of obsessive symptomatology in AN. Antipsychotics like olanzapine and aripiprazole proved to be effective in reducing obsessive thoughts among ED individuals [59]; on the other hand, a recent study found that antidepressants are associated with one higher level of impulsivity [60]. Psychotherapeutic treatments with proven efficacy for obsessive thoughts among OCD individuals are cognitive–behavioral therapy [61], mindfulness-based psychotherapy [62], and new approaches combining relational psychoanalysis, mentalization-based therapy, and mindfulness-based techniques [63]. When applied to AN, psychological and pharmacological treatments may benefit from acknowledging the role of cognitive impulsivity associated with obsessive thoughts and behaviors in AN. 

## Figures and Tables

**Table 1 nutrients-16-01156-t001:** Differences in impulsivity dimensions between AN individuals and HCs.

	AN (n = 53) Mean (SD)	HCs (n = 59) Mean (SD)	t	*p*
**Impulsivity** **First-Order Factors**				
BIS-11 Attention	10.5 (2.8)	10.5 (2.4)	−0.032	0.974
BIS-11 Cognitive Instability	**7.4 (2.0)**	**6.0 (1.9)**	**−3.749**	**<0.001**
BIS-11 Motor Impulsiveness	12.9 (3.6)	13.5 (4.5)	0.804	0.423
BIS-11 Perseverance	7.6 (1.7)	7.6 (1.7)	−0.027	0.978
BIS-11 Self-control	16.8 (3.7)	17.1 (3.6)	0.496	0.621
BIS-11 Cognitive Complexity	12.1 (2.3)	12.7 (2.6)	1.319	0.190
**Impulsivity** **Second-Order Factors**				
BIS-11 Attentional Impulsiveness	17.9 (3.9)	16.5 (3.8)	−1.947	0.054
BIS-11 Motor Impulsiveness	20.5 (4.3)	21.1 (5.3)	0.643	0.522
BIS-11 Non-planning Impulsiveness	28.8 (5.0)	29.8 (5.6)	0.959	0.340
**Impulsivity** **Global Score**				
BIS-11 Global Score	67.2 (9.6)	67.3 (12.6)	0.062	0.950

Legend: AN = anorexia nervosa; HCs = healthy controls; SD = standard deviation; BIS-11 = Barratt Impulsiveness Scale version 11.

**Table 2 nutrients-16-01156-t002:** Pearson’s correlations of total impulsivity and cognitive instability in AN individuals (N = 53).

	BIS-11 Global Score		BIS-11 Cognitive Instability	
	Pearson’s Coefficient	*p*	Pearson’s Coefficient	*p*
Age	−0.183	0.194	−0.303	0.029
BMI	0.071	0.615	0.138	0.329
Years of illness	−0.177	0.213	−0.342	0.014
OCI Washing	0.014	0.922	0.285	0.039
OCI Checking	−0.035	0.803	0.284	0.039
OCI Doubting	0.083	0.555	0.325	0.018
OCI Ordering	0.041	0.771	0.312	0.023
OCI Obsessing	0.404	0.003	**0.594**	**<0.001**
OCI Hoarding	0.006	0.968	0.331	0.016
OCI Mental Neutralizing	0.027	0.846	0.402	0.003
OCI Global Score	0.122	0.389	**0.449**	**<0.001**
STAI State Anxiety	0.157	0.281	0.271	0.060
STAI Trait Anxiety	0.183	0.208	0.337	0.018
EDI–2 Drive for Thinness	0.217	0.152	0.269	0.074
EDI–2 Bulimia	0.443	0.002	0.255	0.091
EDI–2 Body Dissatisfaction	0.107	0.483	0.154	0.311
EDI–2 Ineffectiveness	0.325	0.029	0.336	0.024
EDI–2 Perfectionism	−0.071	0.642	0.179	0.239
EDI–2 Interpersonal Distrust	0.211	0.165	0.329	0.027
EDI–2 Interoceptive Awareness	0.302	0.044	**0.514**	**<0.001**
EDI–2 Maturity Fears	0.152	0.320	0.264	0.079
EDI–2 Asceticism	0.091	0.553	0.272	0.071
EDI–2 Impulse Regulation	0.320	0.032	**0.532**	**<0.001**
EDI–2 Social Insecurity	0.147	0.336	0.335	0.024
BDI	0.441	0.003	0.417	0.005

Legend: AN = anorexia nervosa; BIS-11 = Barratt Impulsiveness Scale version 11; BMI = body mass index; OCI = Obsessive–Compulsive Inventory; STAI = State–Trait Anxiety Inventory; EDI–2 = Eating Disorders Inventory—2; BDI = Beck Depression Inventory.

**Table 3 nutrients-16-01156-t003:** Linear regression model for obsessive thoughts and behaviors (OCI total score) in AN individuals (N = 53).

	Multivariate Regression			Properties of the Model		
Variables	Beta	t	*p*	R	F	*p*
BIS-11 Cognitive Instability	**0.553**	**3.487**	**0.001**	**0.455**	**3.065**	**0.009**
EDI–2 Body Dissatisfaction	**0.641**	**2.812**	**0.008**			
EDI–2 Bulimia	**−0.340**	**−2.263**	**0.030**			
EDI–2 Drive for Thinness	−0.412	−1.835	0.075			
STAI State Anxiety	0.190	1.193	0.241			
BDI	−0.171	−1.035	0.308			
BMI	0.136	0.945	0.351			
STAI Trait Anxiety	0.097	0.515	0.610			
Age	0.058	0.360	0.721			

Legend: AN = anorexia nervosa; BIS-11 = Barratt Impulsiveness Scale version 11; BMI = body mass index; OCI = Obsessive–Compulsive Inventory; STAI = State–Trait Anxiety Inventory; EDI–2 = Eating Disorders Inventory—2; BDI = Beck Depression Inventory.

## Data Availability

The corresponding author havs full access to all the data in the study and takes responsibility for the integrity of the data and the accuracy of data analysis. Data are contained within the article and Appendix A.

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
