# Peer review of "Cognitive Impulsivity in Anorexia Nervosa in Correlation with Eating and Obsessive Symptoms: A Comparison with Healthy Controls"

_nutrients, 2024, doi:10.3390/nu16081156_

Round 1

Reviewer 1 Report

Comments and Suggestions for Authors

Review of nutrients-2866009

This study used a medium-sized AN sample and a control sample to investigate facets of impulsivity against ED symptoms, obsessive-compulsive traits, and other psychopathology. Although the aim to dissect various aspects of cognitive and psychological features in ED is laudable, several factors detract from my enthusiasm in this case, noted below in the approximate order they appear, mixing more and less serious concerns.

Line 60 "Two recent studies have confirmed this trend." – no, the studies cited did not study whether such a trend exists, so this could be rephrased. 

Line 70 "…trauma has increased in importance as a risk factor for EDs" – I would say this is not possible to conclude from two relatively small studies on AN inpatients from a single clinic. I have the overall impression that trauma is less in focus than it used to be.

I suggest using "disorder" rather than "disease" for EDs, OCD etc., it is more correct.

Lines 93-96 "In this perspective, the obsessive thoughts typical of AN-R might represent silent expressions of impulsivity followed by inhibitory control. In these cases, the inhibition of impulsivity is indicative of its active dysregulation rather than its absence." – here is a central theoretical contention in the study that needs to be carefully introduced and supported by literature (see more below). Here, it sounds like the authors are assuming that impulsivity is present because there is no evidence of it, and they assume the mechanism of inhibition to explain this. These assumptions sound counterintuitive and need thorough theoretical and empirical justification. 

The next sentence tries to substantiate the claims: "In fact, the impulsivity inhibition behavior is particularly marked during the acute and most distressful phase of AN, while partially regressing with the recovery (Bardone-Cone et al., 2016)." However, I think that study kind of says the opposite, that negative urgency was most elevated during active ED and regressed with recovery. Negative urgency is not a measure of ”impulsivity inhibition”, is it? It’s a tendency toward impulsive behavior. Also, the majority of their actively ill sample had EDNOS, not AN.

Lines 117-120 "We assume that, following the description of impulsivity in ED in Bardone-Cone et al. (2016) and Lavender and Mitchell (2015), it is possible to describe AN as characterized by increased impulsivity levels concealed by obsessive thoughts and behaviors and inhibition processes" – as noted, I do not think that Bardone-Cone et al showed this, and the Lavender & Mitchell 2015 study is cited here and in other places as support for a silent/inhibited type of impulsivity alongside an explicit one. Could the authors explicate where and how Lavender & Mitchell discuss this? I can’t really find this line of reasoning in their text.

Lines 127-132 "… we expect that AN individuals present increased impulsivity levels compared to healthy controls, that impulsivity is associated with bulimic symptomatology, and that impulsivity predicts a higher presence of obsessive thoughts and behaviors. Further, we expect that in AN, cognitive rather than different dimensions of impulsivity have higher scores and that restricting eating behavior is associated with cognitive impulsivity."; the authors should note that as written, only the expectation that cognitive rather than other impulsivity dimensions was scored higher than in controls received support. It was not general impulsivity that was scored higher or predicted obsessionality, and restricting eating behavior was not measured.

Lines 204-207 – why were these variables selected for the multiple regression?

Line 212 "(mean AN = 6.0)" – 6.0 what? Years? And what is the SD?

Some kind of mass significance correction is needed, there are ca 50 significance tests in a pretty small sample (my suggestion is Bonferroni-Holm), which will change results quite a bit I suspect.

In the Results, p-values are in the tables and do not need to be repeated in the text.

Lines 300-301 "Notably, the latter type of impulsivity [I assume that this is the 'silent, inhibited' kind], in AN, seems to be of cognitive origin and correlated to ED characteristics as measured by EDI-2." - If this refers to BIS cognitive instability, there were no significant correlations with the three EDI-2 ED symptom scales, only associated features that do not explicitly concern ED cognitions/behaviors but that are theoretically assumed to relate to ED. So I think this conclusion needs to be tempered and considered more carefully in terms of what the theoretical implications are. Further, at a guess, if adequate mass significance correction is undertaken, only interoceptive awareness and impulse regulation would remain significant, besides OCI Obsessing and Global score. Interpreting this pattern as centrally relevant to ED symptoms may be a stretch.

Lines 310-314 "The finding of an association of cognitive impulsivity with interpersonal distrust and social insecurity may indicate the existence of mental processes characterized by cognitive impulsivity (and subsequently targeted by inhibition mechanisms), mainly connected to insecurity and difficulties in interpersonal relationships." – doesn't this sentence say the same thing twice? That cognitive impulsivity is associated with interpersonal problems (although with the undemonstrated and unsupported "inhibition mechanisms" added)? 

Lines 326-331 "Moreover, negative cognitive appraisals amplify the emotional pain of obsessions (Salkovskis, 1985). This is particularly interesting since eating and weight concerns are often considered expressions of obsessionality declined into the ED symptomatology, but even of an obsessive personality (Halmi, 2004; Longo et al., 2022; Young et al., 2013). Cognitive impulsivity might thus be the common factor between these two psychopathologies, obsessionality, and eating disorder symptomatology." - This is unclear; what negative cognitive appraisals might this be for example? Appraisals of what? And what does ”declined into the ED” mean? And ending with ”eating disorder symptomatology” is questionable since cognitive impulsivity did not correlate with Bulimia on the EDI-2.

Line 336 "Other significant yet minor associations…" - The table says ”Beta” which is a standardized coefficient, which should mean that the association with body dissatisfaction is stronger?

Line 342, the authors cite the 2022 DSM-5, but this is not a research reference; is there empirical support in the literature for this claim?

Lines 343-344 "… our study supports the hypothesis that obsessive thoughts and behaviors are a silent expression of impulsivity in eating disorders" - I do not see evidence for this ”silent expression”, i.e. the inhibition processes have not been shown empirically, which the authors acknowledge on Line 359. So it looks like the authors agree with me that their own conclusion is unsupported...

A major limitation is that the BIS-11 factor structure is in serious question: several studies have found that its putative 3 factors and 6 primary factors are not statistically supported (one example is Reise, S. P., Moore, T. M., Sabb, F. W., Brown, A. K., & London, E. D. (2013). The Barratt Impulsiveness Scale-11: Reassessment of Its Structure in a Community Sample. Psychological Assessment, 25(2), 631–642. https://doi.org/10.1037/a0032161), and other factor structures are instead suggested. This literature should be taken into consideration, possibly resulting in recalculation of scores and new analyses, or if the present scale use is retained, this should be discussed and justified thoroughly.

Also, it would have made sense to use the control group for more analyses, such as investigating whether similar associations between measures obtain in the group as well. This, besides comparing with other EDs, is needed to justify any claims that results have relevance for AN specifically.

Comments on the Quality of English Language

The manuscript needs proofreading by a native or highly proficient English speaker; there are errors and atypical word and phrase choices that detract from readability. 

Author Response

This study used a medium-sized AN sample and a control sample to investigate facets of impulsivity against ED symptoms, obsessive-compulsive traits, and other psychopathology. Although the aim to dissect various aspects of cognitive and psychological features in ED is laudable, several factors detract from my enthusiasm in this case, noted below in the approximate order they appear, mixing more and less serious concerns.

Answer: We thank you for all your insightful comments. We have extensively revised the paper. Specifically regarding the theoric introduction and the discussion we have cancelled any reference to the silence impulsivity and we have focused the discussion on the analogies between AN and OCD. You can read in the abstract, the introduction, the discussion and the conclusion all the editing we have done.

Line 60 "Two recent studies have confirmed this trend." – no, the studies cited did not study whether such a trend exists, so this could be rephrased. 

Answer: the sentence has been rephrased

Line 70 "…trauma has increased in importance as a risk factor for EDs" – I would say this is not possible to conclude from two relatively small studies on AN inpatients from a single clinic. I have the overall impression that trauma is less in focus than it used to be.

Answer: the complete sentence is cancelled. Trauma is not the focus of our paper.

I suggest using "disorder" rather than "disease" for EDs, OCD etc., it is more correct.

Answer: edited

Lines 93-96 "In this perspective, the obsessive thoughts typical of AN-R might represent silent expressions of impulsivity followed by inhibitory control. In these cases, the inhibition of impulsivity is indicative of its active dysregulation rather than its absence." – here is a central theoretical contention in the study that needs to be carefully introduced and supported by literature (see more below). Here, it sounds like the authors are assuming that impulsivity is present because there is no evidence of it, and they assume the mechanism of inhibition to explain this. These assumptions sound counterintuitive and need thorough theoretical and empirical justification. 

Answer: as we have stated now this whole part of the paper has been reformulated.

The next sentence tries to substantiate the claims: "In fact, the impulsivity inhibition behavior is particularly marked during the acute and most distressful phase of AN, while partially regressing with the recovery (Bardone-Cone et al., 2016)." However, I think that study kind of says the opposite, that negative urgency was most elevated during active ED and regressed with recovery. Negative urgency is not a measure of ”impulsivity inhibition”, is it? It’s a tendency toward impulsive behavior. Also, the majority of their actively ill sample had EDNOS, not AN.

Answer: correct. We have rewritten this paragraph.

Lines 117-120 "We assume that, following the description of impulsivity in ED in Bardone-Cone et al. (2016) and Lavender and Mitchell (2015), it is possible to describe AN as characterized by increased impulsivity levels concealed by obsessive thoughts and behaviors and inhibition processes" – as noted, I do not think that Bardone-Cone et al showed this, and the Lavender & Mitchell 2015 study is cited here and in other places as support for a silent/inhibited type of impulsivity alongside an explicit one. Could the authors explicate where and how Lavender & Mitchell discuss this? I can’t really find this line of reasoning in their text.

Answer: Thanks to your comments we have avoided to use Bardone Cone and Lavender paper. We have acknowledged that their papers did not entail the past hypothesis of our paper and now the introduction is focused on analogy between ocd and an concerning the role of impulsivity

Lines 127-132 "… we expect that AN individuals present increased impulsivity levels compared to healthy controls, that impulsivity is associated with bulimic symptomatology, and that impulsivity predicts a higher presence of obsessive thoughts and behaviors. Further, we expect that in AN, cognitive rather than different dimensions of impulsivity have higher scores and that restricting eating behavior is associated with cognitive impulsivity."; the authors should note that as written, only the expectation that cognitive rather than other impulsivity dimensions was scored higher than in controls received support. It was not general impulsivity that was scored higher or predicted obsessionality, and restricting eating behavior was not measured.

Answer: we have rewritten this paragraph following your suggestions.

Lines 204-207 – why were these variables selected for the multiple regression?

Answer: in the statistical section we have answered this issue.

Line 212 "(mean AN = 6.0)" – 6.0 what? Years? And what is the SD?

Answer: Years of illness. Correction done.

Some kind of mass significance correction is needed, there are ca 50 significance tests in a pretty small sample (my suggestion is Bonferroni-Holm), which will change results quite a bit I suspect.

Answer: We have presented in the data the Bonferroni-holm correction and thus changed the discussion of the results

In the Results, p-values are in the tables and do not need to be repeated in the text.

Answer: ok, done.

Lines 300-301 "Notably, the latter type of impulsivity [I assume that this is the 'silent, inhibited' kind], in AN, seems to be of cognitive origin and correlated to ED characteristics as measured by EDI-2." - If this refers to BIS cognitive instability, there were no significant correlations with the three EDI-2 ED symptom scales, only associated features that do not explicitly concern ED cognitions/behaviors but that are theoretically assumed to relate to ED. So I think this conclusion needs to be tempered and considered more carefully in terms of what the theoretical implications are. Further, at a guess, if adequate mass significance correction is undertaken, only interoceptive awareness and impulse regulation would remain significant, besides OCI Obsessing and Global score. Interpreting this pattern as centrally relevant to ED symptoms may be a stretch.

Answer: This part of the discussion has been rewritten following your indications.

Lines 310-314 "The finding of an association of cognitive impulsivity with interpersonal distrust and social insecurity may indicate the existence of mental processes characterized by cognitive impulsivity (and subsequently targeted by inhibition mechanisms), mainly connected to insecurity and difficulties in interpersonal relationships." – doesn't this sentence say the same thing twice? That cognitive impulsivity is associated with interpersonal problems (although with the undemonstrated and unsupported "inhibition mechanisms" added)? 

Answer: we have cancelled the “inhibition mechanism” part and reformulated the sentence.

Lines 326-331 "Moreover, negative cognitive appraisals amplify the emotional pain of obsessions (Salkovskis, 1985). This is particularly interesting since eating and weight concerns are often considered expressions of obsessionality declined into the ED symptomatology, but even of an obsessive personality (Halmi, 2004; Longo et al., 2022; Young et al., 2013). Cognitive impulsivity might thus be the common factor between these two psychopathologies, obsessionality, and eating disorder symptomatology." - This is unclear; what negative cognitive appraisals might this be for example? Appraisals of what? And what does ”declined into the ED” mean? And ending with ”eating disorder symptomatology” is questionable since cognitive impulsivity did not correlate with Bulimia on the EDI-2.

Answer: this part has been rewritten avoiding any reference to cognitive appraisal and addressing the more in focus aspects of cognitive impulsivity

Line 336 "Other significant yet minor associations…" - The table says ”Beta” which is a standardized coefficient, which should mean that the association with body dissatisfaction is stronger?

Answer: This sentence has been rewritten.

Line 342, the authors cite the 2022 DSM-5, but this is not a research reference; is there empirical support in the literature for this claim?

Answer: the sentence has been cancelled because was redundant

Lines 343-344 "… our study supports the hypothesis that obsessive thoughts and behaviors are a silent expression of impulsivity in eating disorders" - I do not see evidence for this ”silent expression”, i.e. the inhibition processes have not been shown empirically, which the authors acknowledge on Line 359. So it looks like the authors agree with me that their own conclusion is unsupported...

Answer: see previous answers.

A major limitation is that the BIS-11 factor structure is in serious question: several studies have found that its putative 3 factors and 6 primary factors are not statistically supported (one example is Reise, S. P., Moore, T. M., Sabb, F. W., Brown, A. K., & London, E. D. (2013). The Barratt Impulsiveness Scale-11: Reassessment of Its Structure in a Community Sample. Psychological Assessment, 25(2), 631–642. https://doi.org/10.1037/a0032161), and other factor structures are instead suggested. This literature should be taken into consideration, possibly resulting in recalculation of scores and new analyses, or if the present scale use is retained, this should be discussed and justified thoroughly.

Answer: thanks to your comments this issue has been addressed in the limitation paragraph.

Also, it would have made sense to use the control group for more analyses, such as investigating whether similar associations between measures obtain in the group as well. This, besides comparing with other EDs, is needed to justify any claims that results have relevance for AN specifically.

Answer: The analysis in the control group has been added in a table in the supplementary materials. The data has been commented in the discussion.

Reviewer 2 Report

Comments and Suggestions for Authors

Topic is somewhat interesting, however, there are still various sections of the paper that needs revision.

Typically, 1 sentence does not constitute a paragraph - see line 32 to 34, maybe combine with the next sentences.

Only 1 study that suggest, eg Lavender and Mitchell, 2015; because this is the main premise of your paper, there should be more evidence that focus on impulsivity.

Perhaps, provide in terms of hypothesis the research objectives, or research questions - preferably at the end of the introduction

How about exclusion criteria? Are the instruments in English? or translated?

Did you use control variables? for example age and/or BMI to remove individual differences

What now? provide some practical implications

Comments on the Quality of English Language

the introduction part - 1 sentence, typically does not constitute a paragraph

Author Response

Topic is somewhat interesting, however, there are still various sections of the paper that needs revision.

Answer: Thank you for your comments. We really appreciate the suggestions.

Typically, 1 sentence does not constitute a paragraph - see line 32 to 34, maybe combine with the next sentences.

Answer: the sentence has been rewritten

Only 1 study that suggest, eg Lavender and Mitchell, 2015; because this is the main premise of your paper, there should be more evidence that focus on impulsivity.

Perhaps, provide in terms of hypothesis the research objectives, or research questions - preferably at the end of the introduction

Answer: thanks to your comments and the comments of reviewer 1 this whole part is now completely rewritten in the introduction and discussion and in the abstract.

How about exclusion criteria? Are the instruments in English? or translated?

Answer: this has been now added in the text. All test were used in the Italian version

Did you use control variables? for example age and/or BMI to remove individual differences

Answer: no, we have added this in the limitation section.

What now? provide some practical implications

Answer: a final paragraph has been added to the draft addressing pratical implications.

Round 2

Reviewer 1 Report

Comments and Suggestions for Authors

2nd review of ”Cognitive impulsivity in Anorexia Nervosa in correlation with eating and obsessive symptoms: a comparison with healthy controls”. I thank the authors for addressing my concerns and improving some aspects of the study, but I unfortunately have remaining and new concerns. 

Lines 45-46 (1st sentence) says that EDs “typically” (although there is a typo here) manifest with a chronic or relapsing trajectory; this is not true and the sentence was better before when it said “commonly”. Also, the Balasundaram and Santhanam reference is a bit strange, it seems to be a non-peer-reviewed e-book. I would have expected a review or empirical reference here.

In some places, citation numbers (e.g. “[9]” in lines 69-70) are placed both at the start and end of a sentence. 

Something is wrong on Line 72: “[2,8–11]. (2007)[13,14]”.

Please use the AN abbreviation consistently (e.g., Lines 87, 88, 89 and other places).

Line 97, the citations are on the wrong side of the period.

Questionnaires do not have genders in English (lines 160, 165, 169, and 275).

The text and tables variously mention Cognitive Instability and Cognitive Impulsivity which seem to refer to the same subscale. Please be consistent here. 

Line 245 and Table 3, was the association between Bulimia and OCI Total score significant after Bonferroni-Holm? I would have guessed no.

For completeness, control group associations between the BIS and EDI-2, BDI, and STAI should also be shown in Supplementary to further evaluate whether findings are AN-specific or not (which they do not seem to be based on the results shown). And why not include the multiple regression also?

Lines 254-258 describes some associations as significant that were not so noted in Table 2 (BDI and Mental Neutralizing, which is mentioned again on Lines 271 and 320).

Lines 265-267 “This appears to confirm the hypothesis of a similarity in the role of impulsivity in AN and OCD, as also Hudiburgh and colleagues [50] have described attentional impulsivity and not global impulsivity higher in OCD” is a bit confusing since this was not a hypothesis in the present study, and since AN and OCD samples need to be compared directly to draw this conclusion. 

Line 272-273 and 287 says that Cognitive impulsivity (which I assume refers to Cognitive Instability) was significantly associated with all OCI subscales, but after Bonferroni-Holm only Obsessing and Global remained significant. If one chooses to apply a mass significance correction, it should be done consistently and only results that survive should be described as “statistically significant”.

Lines 275-6, it is worth noting here that this association was not unique to AN but rather, since a similar pattern was found in healthy controls, applies to people in general. 

Lines 282-283 says that general impulsivity was linked to purging behavior but there is no single measure of purging behavior in this study. Say the “Bulimia” subscale since it is not known which aspects of that subscale (i.e. binge eating, purging, both) carried the effect. This paragraph also notes associations that were not significant after Bonferroni-Holm. Again, a decision needs to be made regarding whether to acknowledge such effects or not, i.e. should the new corrected significance level guide interpretations or just mentioned as a marker of “stronger significance” or something similar? Given that statistical significance is (perhaps unfortunately) regarded as the yes/no criterion for considering something an effect at all, most readers are likely to expect that this convention is followed.

The paragraph starting on Line 302 discusses results that were not significant according to Table 2.

Line 306 “This association therefore might not be specific to AN and OCD…”; this study did not concern OCD and thus cannot speak to that disorder.

Lines 311-2 describes a hypothesis that cognitive impulsivity was the principle predictor of obsessive thoughts in the AN sample, but I see no such hypothesis in the Intro: “Based on previous findings in the literature, we expect that […] impulsivity predicts a higher presence of obsessive thoughts and behaviors.” Thus, general impulsivity should be associated with impulsive thoughts and behaviors, but this was not found, right? Also, the emphasis on “in the AN sample” seems to suggest that this applies to AN specifically as compared to non-AN, but the regression was not run in healthy controls; this is another reason why it should be included in Supplementary.

A citation is missing on Line 326.

Comments on the Quality of English Language

As noted last time, the study really needs more proofreading, both in recently edited and unedited sections, and there are mistakes with punctuation and citations (see some examples in my comments, and the title is awkward language-wise too).

Author Response

First of all, we would like to thank the reviewers enormously for their careful analysis of the text and the corrections that were helpful in improving our work. We have tried to follow all directions and are sending you the new version of the paper.

Reviewer 1

2nd review of ”Cognitive impulsivity in Anorexia Nervosa in correlation with eating and obsessive symptoms: a comparison with healthy controls”. I thank the authors for addressing my concerns and improving some aspects of the study, but I unfortunately have remaining and new concerns. 

Lines 45-46 (1st sentence) says that EDs “typically” (although there is a typo here) manifest with a chronic or relapsing trajectory; this is not true and the sentence was better before when it said “commonly”. Also, the Balasundaram and Santhanam reference is a bit strange, it seems to be a non-peer-reviewed e-book. I would have expected a review or empirical reference here.

Answer: we have corrected the sentence and the references

In some places, citation numbers (e.g. “[9]” in lines 69-70) are placed both at the start and end of a sentence.  

Something is wrong on Line 72: “[2,8–11]. (2007)[13,14]”.

Answer: we have thoroughly corrected the references in the text

Please use the AN abbreviation consistently (e.g., Lines 87, 88, 89 and other places).

Answer: done

Line 97, the citations are on the wrong side of the period.

Answer: done

Questionnaires do not have genders in English (lines 160, 165, 169, and 275).

Answer: done

The text and tables variously mention Cognitive Instability and Cognitive Impulsivity which seem to refer to the same subscale. Please be consistent here. 

Answer: we have now always used extensively cognitive instability instead of cognitive impulsivity

Line 245 and Table 3, was the association between Bulimia and OCI Total score significant after Bonferroni-Holm? I would have guessed no.

Answer: That is correct. We have edited that part.

For completeness, control group associations between the BIS and EDI-2, BDI, and STAI should also be shown in Supplementary to further evaluate whether findings are AN-specific or not (which they do not seem to be based on the results shown). And why not include the multiple regression also?

Answer: we have included the multiple regression and the correlation of BDI, STAI with BIS TOT e BIS cognitive instability. Since the control group did not have an ED we avoided to add the EDI-2 results in the control group, but it can be available on request.

Lines 254-258 describes some associations as significant that were not so noted in Table 2 (BDI and Mental Neutralizing, which is mentioned again on Lines 271 and 320).

Answer:  the whole result and discussion part has been rewritten in order to follow this and the following comments.

Lines 265-267 “This appears to confirm the hypothesis of a similarity in the role of impulsivity in AN and OCD, as also Hudiburgh and colleagues [50] have described attentional impulsivity and not global impulsivity higher in OCD” is a bit confusing since this was not a hypothesis in the present study, and since AN and OCD samples need to be compared directly to draw this conclusion. 

Answer. This sentence has been cancelled and reformulated.

Line 272-273 and 287 says that Cognitive impulsivity (which I assume refers to Cognitive Instability) was significantly associated with all OCI subscales, but after Bonferroni-Holm only Obsessing and Global remained significant. If one chooses to apply a mass significance correction, it should be done consistently and only results that survive should be described as “statistically significant”.

Answer: see above

Lines 275-6, it is worth noting here that this association was not unique to AN but rather, since a similar pattern was found in healthy controls, applies to people in general. 

Answer: this has been made clear trough the text.

Lines 282-283 says that general impulsivity was linked to purging behavior but there is no single measure of purging behavior in this study. Say the “Bulimia” subscale since it is not known which aspects of that subscale (i.e. binge eating, purging, both) carried the effect. This paragraph also notes associations that were not significant after Bonferroni-Holm. Again, a decision needs to be made regarding whether to acknowledge such effects or not, i.e. should the new corrected significance level guide interpretations or just mentioned as a marker of “stronger significance” or something similar? Given that statistical significance is (perhaps unfortunately) regarded as the yes/no criterion for considering something an effect at all, most readers are likely to expect that this convention is followed.

Answer: this has been corrected. The whole discussion was reformulated and we avoided expressions such as stronger significance.

The paragraph starting on Line 302 discusses results that were not significant according to Table 2.

Answer: done.

Line 306 “This association therefore might not be specific to AN and OCD…”; this study did not concern OCD and thus cannot speak to that disorder.

Answer: this concept has been rewritten following your suggestion.

Lines 311-2 describes a hypothesis that cognitive impulsivity was the principle predictor of obsessive thoughts in the AN sample, but I see no such hypothesis in the Intro: “Based on previous findings in the literature, we expect that […] impulsivity predicts a higher presence of obsessive thoughts and behaviors.” Thus, general impulsivity should be associated with impulsive thoughts and behaviors, but this was not found, right? Also, the emphasis on “in the AN sample” seems to suggest that this applies to AN specifically as compared to non-AN, but the regression was not run in healthy controls; this is another reason why it should be included in Supplementary.

Answer: correct, that was not our first hypothesis. We have rewritten this sentence.

A citation is missing on Line 326.

Answer: citation added.

Reviewer 2 Report

Comments and Suggestions for Authors

At first, there is a concern on the similarity result of 31%, however, upon further checking of the ithenticate results seems fine, the majority of the similarity results are the words native to the paper template, and most parts of the definition and research instrument

it would be advisable that the author/s go in and do some more revisions or rephrasing and try to remove the parts of the template and re-run the similarity check

Author Response

At first, there is a concern on the similarity result of 31%, however, upon further checking of the ithenticate results seems fine, the majority of the similarity results are the words native to the paper template, and most parts of the definition and research instrument

it would be advisable that the author/s go in and do some more revisions or rephrasing and try to remove the parts of the template and re-run the similarity check

Thanks to your comments and those of reviewer 1 we have extensively revised the paper also trying to reformulate concept and sentences.